**Data Availability Statement:** The data can be obtained from the CSA website using http://www.statsethiopia.gov.et.

**Funding:** The authors received no specific funding for this work.

# Prevalence and correlates associated with early childbearing among teenage girls in Ethiopia: A multilevel analysis

**Abebe Debu Liga**[1]*, **Adane Erango Boyamo**[2], **Yasin Negash Jabir**[3], **Akalu Banbeta Tereda**[3]

1 Department of Statistics, College of Natural and Computational Sciences, Wolkite University, Wolkite, Ethiopia, 2 Department of Statistics, College of Natural and Computational Sciences, Wachemo University, Hosaena, Ethiopia, 3 Department of Statistics, College of Natural Science, Jimma University, Jimma, Ethiopia

* abe.debu@yahoo.com

## Abstract

### Background

Teenage childbearing remains a significant global health concern, and in nations with limited resources, it is the major cause of newborn and maternal deaths. Early teenage childbearing is still Ethiopia's public health issue. Therefore, the goal of this study was to identify the prevalence and correlates of influencing early childbearing among teenage girls across Ethiopia.

### Methods

We conducted a secondary analysis of cross-sectional data from the 2016 Ethiopian Demographic and Health Survey. A multistage stratified cluster sampling strategy based on the community was used to include the 3,498 participants in total. To determine the significantly correlated factors that influence adolescent pregnancy, a multilevel binary logistic regression analysis was used. The factors that have a significant association with early childbearing were identified using the Adjusted Odds Ratio (AOR) and 95% Confidence Interval (CI).

### Results

This study demonstrated that 10.3% of teens across the country had children at an early age. The odds of early childbearing among teenage girls increased with first marriages occurring before the age of 18, non-formal education, being from a lower- or middle-class family, not using contraceptives, following Muslim or other religious beliefs, and being aware of the fertile window. Teenagers who had exposure to the media, however, had a reduced chance of becoming pregnant early.

### Conclusions

The study indicates that early teenage childbearing is still Ethiopia's most significant public health problem. Therefore, the Ethiopian government should ban early marriage while also taking steps to reduce the risk through formal education, improved access to reproductive

**Competing interests:** The authors declared that no competing interests exist.

**Abbreviations:** AIC, Akaike Information Criterion; BIC, Bayesian Information Criterion; CSA, Central Statistical Agency; EDHS, Ethiopian Demographic and Health Survey; HIV, Human Immune Virus; ICC, Intra-class Correlation Coefficient; OR, Odds Ratio; SSAR, Sub-Saharan African regions; UNICEF, United Nations Children Fund; WHO, World Health Organization.

health education, and contraception, particularly for adolescent girls from low-income families and, by educating religious institutions about pregnancy dangers.

## Introduction

Childbearing among girls between the ages of 10 and 19 is known as teenage pregnancy [1]. The United Nations Children Fund (UNICEF) also describes adolescent pregnancy as "a teenage girl, typically between the ages of 13 and 19, becoming pregnant" [2]. Teenage childbearing remains a serious concern and health issue worldwide, and it is the main cause of newborn and mother death in nations with poor infrastructure [3]. In low- and middle-income countries, pregnancy-related problems are the leading cause of death for girls between the ages of 15 and 19 [4]. These women have more than twice the risk of dying during pregnancy and childbirth compared to women in their twenties, while the risk is five times higher for those under the age of 15 [5]. Teenagers are often inexperienced about the risks of sexual activity and reproductive health [6].

Pregnancy during the teenage years has several of adverse effects, including preterm labor and low birth weight [3, 7], neonatal death, obstructed labor, genital fistula, and eclampsia, as well as the emotional and physical challenges of pregnancy [8–10]. The reproductive health of women is additionally affected by unsafe abortion, sexually transmitted infections like HIV/AIDS, sexual violence, abortion, and restricted access to medical care [11]. Pregnancy among teens seems to be a global problem. Even though rates of teenage pregnancy have declined in most high-income countries since the mid-1990s [12], they remain a substantial public health concern, particularly in Africa [13].

Africa has a higher rate of teenage pregnancies than the rest of the world [14, 15], accounting for 18.8% of all teenage pregnancies, with 19.3% of those occurring in the sub-Saharan region [15]. In seven nations, including Ethiopia, more than 50% of all teenage births occur [16]. In low- and middle-income nations, estimated 21 million and 12 million girls between the ages of 15 and 19 become pregnant and give birth every year, respectively [15]. Studies have shown that reducing the high rate of adolescent pregnancy in developing countries can lower the high level of morbidity and mortality among mothers and newborns [17].

Nevertheless, the Ethiopian government has made reducing the risk of teen pregnancy one of its main priorities for sustainable development [18]. However, Ethiopia has a comparable situation to other sub-Saharan countries in terms of the prevalence of teenage childbearing. Approximately 16% of teenage pregnancies occurred, according to the EDHS-2016 report, with those living in rural areas experiencing a greater pregnancy rate [19]. Additionally, Mamo et al. predicted that the prevalence of teenage pregnancies in Ethiopia would be 23.59% in 2021 [20]. In 2011, a WHO study found that teenage girls who fall pregnant before the age of 18 are 60% more likely to lose their child during the first few years of life [21] and are substantially more likely to face domestic violence [22]. Previous studies identified that early sexual activity [23, 24], early marriage [25], marital status [18, 26, 29], education attainment [27–30], religion [31, 32, 47], age [18, 26, 29, 31, 33], place of residence [29, 31, 33, 34], household wealth [23, 24, 28], employment [26, 29], media exposure [30, 31, 35, 36], alcohol consumption [37], rape [13], contraceptive use [18, 23, 24, 29, 31, 33], lack of comprehensive sexuality education [38], and influence or pressure from peers [31] are all significantly associated with correlates of teenage pregnancy. In Ethiopia, the main obstacles preventing teenagers from using contraceptives were expected stigmas brought on by social norms and attitudes regarding

contraceptive methods, including fear of being observed by others and shame in finding contraceptive methods [39].

Although early marriage has adverse implications, however it is still prevalent in Ethiopia, most parents do not want to marry off their daughters before the age of eighteen [40]. According to Ethiopia's amended family code, getting married before reaching 18 is no longer acceptable [41]. Contrarily, in rural Ethiopia, parents make the majority of marriage-related decisions on behalf of their teenage daughters, including who and when to marry [42]. Among the most frequently claimed arguments in favor of early marriage is the desire to raise the family's standard of living [43].

Numerous studies have revealed a variety of factors that affect teen pregnancy both in Ethiopia and other countries, as identified earlier [4, 13, 15, 18, 20, 33, and 34]. The majority of the previously conducted studies are small-scale, institutional-based, or specific to some area; they lack national-level information. These researchers only used analytical techniques that presumed the independence of each observation, and they only looked at the effects of individual-level traits. The individual observations do, however, exhibit some association within the cluster or group to which they belong due to shared traits [44]. Thus, neglecting this information typically leads to inaccurate conclusions about how attributed factors affect teenage pregnancy [45]. Therefore, the current study handles random effects, regional variation, and subject-specific variance among the correlates of early teenage childbearing using multilevel models to address these issues. Moreover, the finding serves to aid policymakers, program administrators, and healthcare authorities in making better planning and problem-solving decisions.

## Methods

### Data source and sampling procedure

The present study utilized the data of the 2016 Ethiopian Demographic and Health Survey, which had been carried out in Ethiopia between January 18 and June 27, 2016, by the Central Statistical Agency. The survey has been carried out all over Ethiopia utilizing the community-based multistage stratified cluster sampling approach with two stages, including 645 counting areas. The study's sampled locations were split into two groups: 28 households in each cluster in the second phase and, 202 urban and 443 rural areas in the first phase. The interview covered all females between the ages of 15 and 49. The survey covered 16,583 women in total, 3,498 of whom provided their information for the present study [19].

### Study variables

The status of early childbearing among teenage girls was the response variable used in this study, and it was coded as "1" if the teenager became pregnant at an early age and "0" when they did not. While age at first marriage, residence, educational attainment, region, household wealth, knowledge of the fertility window, mass media exposure (watching television, listening to the radio, and reading a newspaper or magazine at least once a week), employment status, religion, contraceptive use, and alcohol consumption were all independent variables taken into consideration for this study.

### Inclusion and exclusion criteria

Teenage girls under the age of nineteen who had complete information regarding early childbearing at the period of collecting data were included, while those under the age of 19 who were lacking in this information weren't included in the study.

## Statistical analysis

Each variable is presented along with its absolute and relative frequencies. They were cross-tabulated according to whether teenage girls had children early (yes or no), and the relationship to each independent variable was examined using chi-square tests. In these univariate analyses, variables that were significant at the 5% level were included in a multilevel logistic regression model with the status of early childbearing among teenage girls as the outcome. 95% confidence intervals and odds ratios have been established for the explanatory variables. All analyses were conducted using SPSS 21.0 (IBM Corp., Armonk, NY) and STATA 23 version.

## Multilevel logistic regression model

A multilevel logistic regression model is an extension of a single-level logistic regression model that includes random effects in studies with nested data structured by more than one level [46]. Individuals at a lower level that are nested within integrated units at a higher level frequently serve as the focus of the analysis. Teenagers were taken as level 1 in this study's two-level logistic regression model, whereas regions were considered level 2 to identify unexplained variance both within and between groups. In this study, three multilevel logistic regression models a null model (model without predictors) or no explanatory variable which assessed the overall variance in the factors affecting childbearing within regions were examined hierarchically. Only variables at the individual level were considered in Model 2. The combined effects of variables at the individual and regional levels were assessed by Model-3.

## Model comparison, adequacy, and heterogeneity checking

The multilevel logistic regression model must first be determined to be appropriate for the data set. Therefore, we applied the standard chi-square test for the contingency to determine whether or not there is systematic variation between the groups or regions. The most common techniques for assessing the degree of accuracy of fit in multilevel logistic regression models were deviance, AIC, and BIC values after it had been demonstrated that the data were appropriate for the model. The null and random coefficient models' deviation, AIC, and BIC scores provide the best model for predicting the prevalence of early teenage childbearing. Additionally, the intra-class correlation coefficient (ICC) quantifies the percentage of variance in the outcome variable that can be accounted for by the grouping structure. The clustered nature of the data and variation within and across communities were taken into account by assuming that each community possessed a distinct intercept and fixed coefficient.

## Ethics approval

Secondary data sets from the 2016 EDHS, which were previously collected and held in confidence, were utilized in this analysis. Therefore, since we did not directly gather data from individuals, the recruitment and involvement of participants were not necessary. The datasets can be accessed with the permission of the Measure DHS program and the Ethiopian Public Health Institute.

# Results

## Descriptive statistics results

This study involved 3498 teenagers from nine regional states and two administrative cities in Ethiopia. Among them, 359 (10.3%) teenage girls gave birth early during the period of gathering the data. Of these, 94 (2.7%) teens became pregnant before the age of 15, and 265 (7.6%)

were pregnant between the ages of 15 and 19. The highest prevalence of teen girls who became pregnant early was found in Affar (20.68%), Gambela (15.57%), and Somali (15.36%), as opposed to Addis Ababa (1.85%). Also, majorities (64.38%) were from rural areas, and 41.02% followed Muslim faiths. Out of all the teenage girls, 16.29% had no formal education at all, 57.95% had completed elementary school, 22.81% had attended secondary school, and 2.94% had attended higher education. Similarly, 19.01 percent, 34.59 percent, and 46.40 percent of teenagers had been married for the first time when they were aged 15 to 17, 15 to 17, and older than 18, respectively (Table 1).

According to the summary, 34.42% of teenagers come from low-income families, followed by 13.44% from middle-class families, and 52.14% from wealthy households. A majority of teenagers (28.53%) were not aware of when they were most fertile; 93.97% of them did not use contraceptives; 74.99% were unemployed, 55.63% were exposed to the mass media, and 97.71 did not use alcohol (Table 1).

Associations with the status of childbearing among teenage girls were significant in univariate analysis for the following variables, age at first marriage, region, place of residence, educational attainment, household wealth, knowledge of the fertile window, contraceptive use, religion, alcohol consumption, and mass media exposure (Table 1).

## Multilevel logistic regression results

**Tests of heterogeneity.** The chi-square test of heterogeneity value in Table 2 is 101.880 with a p-value less than 0.05, which is significant and indicates that there is heterogeneity between childbearing among teenage girls in the region of Ethiopia. It is therefore appropriate to examine a data set using multilevel logistic regression.

**Results of multilevel binary logistic regression model.** The deviance, Akaike information criterion (AIC), and Bayesian information criterion (BIC) values were used to select the best multilevel models among the three fitted models. A model with the lowest AIC and deviance scores would be the best one. Therefore, the random slope model (Model 3) has lower deviance and AIC values than both the empty model (Model 1) and the random intercept model (Model 2), suggesting that it is more accurate in predicting early childbearing girls in Ethiopia (Table 3). Educational attainment and use of contraceptives were the two factors that varied the most throughout the region compared to other variables and were suitable for analysis utilizing the multilevel random slope model. The null model from Table 3 has an ICC of 0.1016, indicating that the grouping structure in higher-level units or regions can account for 10.16% of the variation in the prevalence of early childbearing and 89.94% of the variation is explained within lower-level units.

Table 4 also revealed that there is significant heterogeneity or variance among the teen childbearing rates in the regional states of Ethiopia, with the confidence interval of the estimate of the random intercept excluding zero. Furthermore, the confidence interval for the random slopes for contraceptive use and educational attainment excludes zero, demonstrating that the educational and contraceptive practices used by teenagers vary from region to region.

The random slope multilevel model reveals age at first marriage, education attainment, household wealth, contraceptive use, knowledge of the fertile window, religion, and exposure to media as significantly (p<0.05) associated with early childbearing among teenage girls (Table 5).

Early marriage has been significantly associated with higher rates of early childbearing. Responding individuals with a first marriage that was less than 15 years old had an odds ratio of 3.14 (95% CI: 2.39–4.14) compared to the reference group with a first marriage that was greater than or equal to 18 years old, and participants with a first marriage that was between 15 and 17 years old had an odds ratio of 2.23 (95% CI: 1.59–3.13).

**Table 1. Socio-demographic characteristics of childbearing status among teenage girls.**

| Variables | Categories | Counts (%) | Early childbearing status | | Chi-square test | | |
|---|---|---|---|---|---|---|---|
| | | | No (89.7%) | Yes (10.3%) | Value | df | P-value |
| Region | Tigray | 423 (12.09) | 388 | 35 (8.27) | 101.88 | 10 | <0.0001 |
| | Afar | 266 (7.60) | 211 | 55 (20.68) | | | |
| | Amhara | 355 (10.15) | 330 | 25 (7.04) | | | |
| | Oromia | 415 (11.86) | 358 | 57 (13.73) | | | |
| | Somali | 319 (9.12) | 270 | 49 (15.36) | | | |
| | Benishangul | 237 (6.78) | 212 | 25 (10.55) | | | |
| | SNNPR | 391 (11.18) | 364 | 27 (6.91) | | | |
| | Gambela | 212 (6.06) | 179 | 33 (15.57) | | | |
| | Harari | 183 (5.23) | 157 | 26 (14.21) | | | |
| | Addis Ababa | 432 (12.35) | 424 | 8 (1.85) | | | |
| | Dire Dawa | 265 (7.58) | 246 | 19 (7.17) | | | |
| Place of Residence | Urban | 1246 (35.62) | 1193 | 53 (4.25) | 75.89 | 1 | <0.0001 |
| | Rural | 2252 (64.38) | 1946 | 306 (13.59) | | | |
| Age at first marriage | < 15 years | 665 (19.01) | 571 | 94 (14.13) | 96.42 | 2 | <0.0001 |
| | 15–17 years | 1210 (34.59) | 1024 | 186 (11.46) | | | |
| | ≥ 18 years | 1623 (46.40) | 1544 | 79 (4.87) | | | |
| Religion | Orthodox | 1409 (40.28) | 1335 | 74 (5.25) | 71.499 | 2 | <0.0001 |
| | Muslim | 1435 (41.02) | 1222 | 213 (14.84) | | | |
| | Others | 654 (18.70) | 582 | 72 (11.01) | | | |
| Education | No education | 570 (16.29) | 442 | 128 (22.46) | 128.57 | 3 | <0.0001 |
| | Primary | 2027 (57.95) | 1835 | 192 (9.47) | | | |
| | Secondary | 798 (22.81) | 761 | 37 (4.64) | | | |
| | Higher | 103 (2.94) | 101 | 2 (1.94) | | | |
| Household wealth | Poor | 1204 (34.42) | 992 | 212 (17.61) | 128.54 | 2 | <0.0001 |
| | Middle | 470 (13.44) | 413 | 57 (12.13) | | | |
| | Rich | 1824 (52.14) | 1734 | 90 (4.93) | | | |
| Knowledge of the fertile window | During period | 147 (4.20) | 132 | 15 (10.20) | 62.20 | 5 | <0.0001 |
| | After period | 656 (18.75) | 537 | 119 (18.14) | | | |
| | Midpoint of period | 648 (18.52) | 592 | 56 (8.64) | | | |
| | Before period | 264 (7.55) | 237 | 27 (10.23) | | | |
| | Every time | 785 (22.44) | 707 | 78 (9.94) | | | |
| | Don't know | 998 (28.53) | 934 | 64 (6.41) | | | |
| Contraceptive uses | None | 3287 (93.97) | 3015 | 272 (8.28) | 233.84 | 1 | <0.0001 |
| | Modern method | 211 (6.03) | 124 | 87 (41.23) | | | |
| Employment Status | No | 2623 (74.99) | 2342 | 281 (10.71) | 2.30 | 1 | 0.129 |
| | Yes | 875 (25.01) | 797 | 78 (8.91) | | | |
| Mass Media Exposure | No | 1552 (44.37) | 1321 | 231 (14.88) | 64.68 | 1 | 0.008 |
| | Yes | 1946 (55.63) | 1818 | 128 (6.58) | | | |
| Alcohol consumption | No | 3418 (97.71) | 3087 | 331 (9.68) | 54.39 | 1 | <0.0001 |
| | Yes | 80 (2.29) | 52 | 28 (35) | | | |

**Table 2. Chi-square tests of heterogeneity.**

| Statistics | Value | Df | p-value |
|---|---|---|---|
| Pearson Chi-Square | 101.88 | 10 | <0.0001 |

**Table 3. Summary results of model selection criteria and ICC.**

| Class | Null model (Model 1) | Random intercept model (Model 2) | Random slope model (Model 3) |
|---|---|---|---|
| Deviance | 72.02(p<0.0001) | 409.7528(p<0.0001) | 30.0107(p<0.0001) |
| AIC | 2246.42 | 1864.67 | 1844.65 |
| BIC | 2258.74 | 1963.23 | 1974.01 |
| ICC | 0.102 | | |

Conversely, higher education significantly decreased the rate of early childbearing. The odds ratio for individuals with at least a primary education was 0.43 (95% CI: 0.28–0.64) compared to the reference group of people with no formal education. Similarly, participants with at least a secondary education had odds ratios of 0.26 (95% CI: 0.12–0.53), and participants with higher education had odds ratios of 0.10 (95% CI: 0.02–0.59), compared to the reference group with no formal education.

Early childbearing and participants' religious affiliation were significantly associated. Muslim religion participants had a higher likelihood of being pregnant earlier compared to Orthodox religious participants (OR: 2.83, 95% CI: 1.92–4.17) and other religious participants had a higher chance of getting pregnant earlier compared to Orthodox religion participants (OR: 2.27, 95% CI: 1.40, 3.69). Similarly, early pregnancy rates were significantly associated with household wealth.

Teen girls from lower household wealth backgrounds were more likely to have children early than those from higher wealth histories (OR: 2.88, 95% CI: 2.03–4.09), and girls from middle household wealth backgrounds were more likely to have children early than those from higher wealth households (OR: 2.33, 95% CI: 1.55–3.52). Participants who had access to mass media had a lower likelihood of practicing early childbearing than those who had no access to it (OR: 0.76, 95% CI: 0.56–1.02). The use of contraceptives greatly reduced the rate of early childbirth.

The use of contraceptives significantly decreased the rate of early childbirth. The odds of early childbearing were higher among those who did not utilize contraceptive methods than among those who did (OR: 18.46, 95% CI: 6.89–49.49). Furthermore, the results suggest that participants' knowledge of the fertile window greatly boosted their likelihood of becoming pregnant early (Table 5).

## Discussion

This study sought to find the prevalence and correlates of factors influencing early childbearing among teenage girls across Ethiopian regions. In this study, 10.3% of participants had children at an early age during the period of gathering the data. The prevalence is higher compared to a study that found 18.8% in Africa and 19.3% in sub-Saharan African nations [15], 28.6 in Wogedi, Northeast Ethiopia, in 2017 [33], and 30.2 in Kersa, Eastern Ethiopia, in

**Table 4. Confidence interval of random intercept and random slope models.**

| Random effects Parameters Region: Unstructured | Estimate | Std. Err. | [95% Conf. Interval] |
|---|---|---|---|
| Variance (contraceptive use) | 2.05 | 1.27 | [0.61 6.91] |
| Variance(Educational attainment) | 0.19 | 0.13 | [0.05 0.75] |
| Variance (cons) | 0.02 | 0.09 | [0.00 0.67] |

**Table 5. Random slope multilevel logistic results of factors associated with teenage childbearing.**

| Variable | Category | AOR | 95% Conf. Interval | P-value |
|---|---|---|---|---|
| Age at first marriage | ≥18 years (Ref.) | | | |
| | 15–17 years | 2.23 | (1.59, 3.13) | <0.0001 |
| | < 15 years | 3.14 | (2.39, 4.14) | <0.0001 |
| Education | No education (Ref.) | | | |
| | Primary | | (0.28, 0.64) | <0.0001 |
| | Secondary | 0.26 | (0.12, 0.53) | <0.0001 |
| | Higher | 0.10 | (0.02, 0.59) | <0.026 |
| Religion | Orthodox (Ref.) | | | |
| | Muslim | 2.83 | (1.92, 4.17) | <0.0001 |
| | Other | 2.27 | (1.40, 3.69) | <0.0001 |
| Household wealth | Rich (Ref.) | | | |
| | Middle | 2.33 | (1.55, 3.52) | <0.0001 |
| | Poor | 2.88 | (2.03, 4.09) | <0.0001 |
| Media Exposure | No (Ref.) | | | |
| | Yes | 0.76 | (0.56, 1.02) | 0.049 |
| Contraceptives use | Yes (Ref.) | | | |
| | No | 18.46 | (6.89, 49.49) | <0.0001 |
| Knowledge of fertile window | Don't know (Ref.) | | | |
| | During period | 2.60 | (1.32, 5.15) | 0.006 |
| | After period | 4.56 | (3.17, 6.56) | <0.0001 |
| | Midpoint of period | 2.90 | (1.89, 4.46) | <0.0001 |
| | Before period | 2.98 | (1.71, 5.20) | <0.0001 |
| | Every time | 2.07 | (1.42, 3.02) | <0.0001 |
| Constant | | 0.02 | (0.01, 0.04) | <0.0001 |

2020 [47]. It might be challenging to provide an appealing rationale for the discrepancy; yet, it could have some relation to the societal and cultural circumstances of the study's locations. There are also regional variations in the rate of early childbearing. Girls from the Afar, Gambella, Somali, and Harari areas had higher rates, while Addis Ababa had the lowest rate across Ethiopia's regions. This is probably due to regional differences in cultural and religious perspectives on sexuality, marriage, and reproduction.

According to this study, girls who got married early had an increased chance of having children. This finding is in line with a study by Presler-Marshall and Jones [25], which indicated that early marriage increased the risk of teenage childbirth, as well as with earlier studies that showed that early sexual activity increased the likelihood of early childbearing [23, 24]. This might be because early marriage leads to multiple births in less than 24 months, repeated unplanned pregnancies, pregnancy terminations, and premature sterility due to a lack of access to contraception. As a result, having children at an early age puts both the mother's and the child's health in danger and increases the likelihood that an infant may pass away in the first year of life [21].

Higher education also significantly reduces the rate of early childbirth. This is in agreement with studies conducted by Habito et al. [27], Baba and Fujiwara [28], Eyasu [29], Worku et al. [30], and Nalenga [31] that found a higher prevalence of teenage births among teenagers with no formal education. Similarly to that, CSA, 2016 revealed that having more education has been associated with a lower risk of being exposed to teen pregnancies [19]. Furthermore, these studies have consistently shown that higher education not only decreases the likelihood

of early childbirth but also provides individuals with the knowledge and resources to make informed decisions about their reproductive health. This suggests that investing in education can be a crucial strategy for reducing teenage pregnancies and promoting overall well-being among young people.

Teenage girls from lower and middle-income families were more likely to get pregnant early than those from higher-income families. The findings fall in line with the studies done by Baba [28], Morón-Duarte [23], and Santos [24], which found that teenage girls who lived with poor parents were more likely to become pregnant early. The possible reason might be that teenagers from low-income societies may not have enough access to knowledge about sexual and reproductive health issues, which increases their risk of having children at an earlier age.

The study found that respondents from the Muslim religion were more likely to become pregnant earlier. This finding is in agreement with a study conducted by Hossain [32], which reported that Muslim women give birth earlier and have more children overall than women with non-Muslim partners. The finding also corresponds with a study conducted by Nalenga [31] and Hossain [48] that found Muslims require more children than Christians. The other study finding reveals that Islamic authorities disapproved of several parts of family planning, particularly abortion and women's liberty [49]. Muslims often have a positive attitude toward having more children and becoming early pregnant due to various reasons, such as the fact that the number of believers in the religion will rise if they have more children, the happiness increases as the family's size grows, a woman's children are her last source of hope if she loses her husband, etc. [48]. Although the findings of this study confirm the Islamic belief that children should be born at an early age, having children before reaching the age of 18 brings a number of health dangers for mothers, including an increased likelihood of pregnancy issues like preterm birth and low birth weight. Therefore, we recommend against giving birth before the age of 18, as it will be detrimental to both the mother's and the child's health.

Teenagers who obtained information from media sources including newspapers, radio, and television were less likely to become pregnant early than those who were not exposed to the media. This finding coincides with that of other studies done by Fatema [35] and Luchuo [36], which found that teenagers who were exposed to mass media had a lower chance of becoming pregnant early. The most likely explanation is that teenage girls are more likely to effectively use maternal health care when they are exposed to a variety of mass media.

This study found that teenage girls who did not use contraceptive methods had a greater probability of having children early. The findings correspond with a study conducted by Morón-Duarte et al. [23], Santos [24], and Ayele et al. [18], which found that teenage childbirth is increased by a lack of contraceptive information and use.

Furthermore, teenagers who were aware of the fertile window during the menstrual cycle had an increased chance of becoming pregnant early. This finding contrasts with a study by Mezmur [47], who found that teenagers who were unaware of the fertile period of the menstrual cycle had a higher prevalence of childbirth.

## Conclusion

The study reveals that 10.3% of teenagers nationwide have children at an early age. The finding indicated that age at first marriage before 18 years, non-formal education, lower- and middle-class household backgrounds, not using contraceptive methods, followers of Muslim and other religions, no access to media, and knowledge of the fertile window were all correlates of early teenage childbearing in the study. Therefore, the government must prioritize the inclusion of comprehensive sex education in school curricula and ensure that all adolescents, regardless of their socioeconomic background, have access to accurate information about contraception

and reproductive health. Religious institutions should also educate their members about the dangers of getting pregnant before turning 18, as doing so is beneficial for both the mother's and the child's health. By addressing these underlying issues, society can work towards reducing early teenage childbearing rates and empowering young girls to make informed choices about their futures.

## Strengths and limitations of the study

The study had several strengths. First, it was based on nationally representative data, and second, it may provide policymakers and program planners with information that would help them develop effective intervention strategies at both the national and regional levels. Thirdly, the EDHS data are hierarchical, so the multilevel analysis was employed to account for this and obtain accurate estimates and standard errors. The EDHS survey was based on respondents' self-reports, which may have given rise to recall bias, which is one weakness of this study. Additionally, because this study was based on secondary data, it is missing several crucial factors that could affect the outcome variable.

## Acknowledgments

We appreciate that the Ethiopian Central Statistical Agency (CSA) provided the EDHS dataset. The authors also expressed gratitude to every expert and research assistant who contributed to the study, as well.

## Author Contributions

**Conceptualization:** Abebe Debu Liga, Adane Erango Boyamo, Yasin Negash Jabir, Akalu Banbeta Tereda.

**Data curation:** Abebe Debu Liga, Adane Erango Boyamo, Yasin Negash Jabir, Akalu Banbeta Tereda.

**Formal analysis:** Abebe Debu Liga, Adane Erango Boyamo, Yasin Negash Jabir, Akalu Banbeta Tereda.

**Investigation:** Abebe Debu Liga, Adane Erango Boyamo, Yasin Negash Jabir, Akalu Banbeta Tereda.

**Methodology:** Abebe Debu Liga, Adane Erango Boyamo, Yasin Negash Jabir, Akalu Banbeta Tereda.

**Project administration:** Abebe Debu Liga.

**Resources:** Abebe Debu Liga, Adane Erango Boyamo, Yasin Negash Jabir, Akalu Banbeta Tereda.

**Software:** Abebe Debu Liga, Adane Erango Boyamo, Yasin Negash Jabir, Akalu Banbeta Tereda.

**Supervision:** Abebe Debu Liga, Adane Erango Boyamo, Yasin Negash Jabir, Akalu Banbeta Tereda.

**Validation:** Abebe Debu Liga, Adane Erango Boyamo, Yasin Negash Jabir, Akalu Banbeta Tereda.

**Visualization:** Abebe Debu Liga, Adane Erango Boyamo, Yasin Negash Jabir, Akalu Banbeta Tereda.

**Writing – original draft:** Abebe Debu Liga.

**Writing – review & editing:** Abebe Debu Liga, Adane Erango Boyamo, Yasin Negash Jabir, Akalu Banbeta Tereda.

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
