## [Decision Letter · Decision Letter 0]

25 May 2023

PONE-D-22-18542Risk factors and prevalence associated with early childbearing among teenage girls in Ethiopia: A Multilevel AnalysisPLOS ONE

Dear Dr. Liga,

Thank you for submitting your manuscript to PLOS ONE. After careful consideration, we feel that it has merit but does not fully meet PLOS ONE’s publication criteria as it currently stands. Therefore, we invite you to submit a revised version of the manuscript that addresses the points raised during the review process.

The manuscript has undergone a thorough review by two reviewers, and their comments are provided below for your reference. The reviewers have recommended English language copy editing to improve the presentation. Additionally, they have requested several clarifications regarding the study methodology to ensure the reproducibility of the reported methods.

We look forward to receiving your revised manuscript.

Kind regards,

Lucinda Shen, MSc

Staff Editor

PLOS ONE

A clean copy of the edited manuscript (uploaded as the new *manuscript* file).

“No funding was obtained for this study.”

Reviewers' comments:

Reviewer's Responses to Questions

**Comments to the Author**

1. Is the manuscript technically sound, and do the data support the conclusions?

Reviewer #1: Yes

Reviewer #2: Partly

2. Has the statistical analysis been performed appropriately and rigorously? 

Reviewer #1: Yes

Reviewer #2: I Don't Know

3. Have the authors made all data underlying the findings in their manuscript fully available?

Reviewer #1: No

Reviewer #2: No

4. Is the manuscript presented in an intelligible fashion and written in standard English?

Reviewer #1: Yes

Reviewer #2: No

5. Review Comments to the Author

Reviewer #1: good job. adolescent reproductive health is always at the center stage because of worsened maternal and newborn health indices. however, you need to describe the ethical processes in data collection or approval to use these secondary data.

Reviewer #2: Authors must consider engaging an English language expert to help undertake a complete language editing of the entire research work; subject verb agreement, grammatical errors, unusually long sentences and to mention a few.

The aim of the study is unclear.

Methodology

Authors should rephrase this statement for clarity--"This study was conducted in 2016 by the Ethiopian Demographic and Health Survey". If secondary data was obtained for this work, it should be clearly stated with appropriate referencing of primary data.

Sections of the methodology indicate male and females ages 15-59 years were interviewed. Were males included in this study. Authors should please clarify.

Authors indicated that all girls aged between 15 to 19 years were included in the study and those out of the range were

excluded. How was consent obtained for females less than 18 years of age? Was accent sought from this age group in the primary data.

Results

There seem to be copious repetition of the results in the tables in the write up. Authors can consider stating just significant ones and leave the rest in the illustrated tables for readers to peruse.

Discussion

In-text referencing of CSA, 2016 should be properly done.

Can authors identify some study limitations and how their study design or approach sought to curtail the limitations.

6. PLOS authors have the option to publish the peer review history of their article (what does this mean?). If published, this will include your full peer review and any attached files.

Reviewer #1: **Yes: **Emmanuel Ugwa, PhD

Reviewer #2: No

---

## [Author Response · Author response to Decision Letter 0]

7 Jun 2023

Author’s response: As suggested by the editor, we made an attempt to keep to the guidelines of the PLOS ONE style.

Author’s response: According to the editorial suggestions, we rewrote the entire document using proper grammar.

Author’s response: The manuscript has been edited by Abebe Debu Liga, and Akalu Banbeta Tereda.

A clean copy of the edited manuscript (uploaded as the new *manuscript* file).

“No funding was obtained for this study.”

Author’s response: Amended it per the editorial comment (Please see the “Financial Disclosure Statement” section)

Author’s response: The data used for this study has been in the hands of the first and corresponding authors. So, it will be sent with a reasonable request.

Author’s response: Amended it per the editorial comment.

Reviewers' comments:

Reviewer's Responses to Questions

Comments to the Author

1. Is the manuscript technically sound, and do the data support the conclusions?

Reviewer #1: Yes

Reviewer #2: Partly

2. Has the statistical analysis been performed appropriately and rigorously?

Reviewer #1: Yes

Reviewer #2: I Don't Know

3. Have the authors made all data underlying the findings in their manuscript fully available?

Reviewer #1: No

Reviewer #2: No

4. Is the manuscript presented in an intelligible fashion and written in standard English?

Reviewer #1: Yes

Reviewer #2: No

Author’s response: We amended the grammar throughout the paper per the reviewer comment.

5. Review Comments to the Author

Reviewer #1: good job. Adolescent reproductive health is always at the center stage because of worsened maternal and newborn health indices. however, you need to describe the ethical processes in data collection or approval to use these secondary data.

Author’s response: At the end of the method section, we added the Ethical Approval Statement to the document.

Reviewer #2: Authors must consider engaging an English language expert to help undertake a complete language editing of the entire research work; subject verb agreement, grammatical errors, unusually long sentences and to mention a few.

Author’s response: According to the reviewers and editor suggestions, we rewrote the entire document using proper grammar.

The aim of the study is unclear.

Author’s response: We attempted to clarify the purpose of this study (please see the third sentence of the abstract's background and the last paragraph in the introduction).

Methodology

Authors should rephrase this statement for clarity--"This study was conducted in 2016 by the Ethiopian Demographic and Health Survey". If secondary data was obtained for this work, it should be clearly stated with appropriate referencing of primary data.

Author’s response: We added the relevant reference for it (please see Data source section in the Methods)

Sections of the methodology indicate male and females ages 15-59 years were interviewed. Were males included in this study? Authors should please clarify.

Author’s response: Actually, only girls under the age of 19 were considered in this study. This study, on the other hand, was based on survey data collected by the Ethiopian Central Statistical Agency. So the data includes males, females, children, and so on. Therefore, we reported the number of males for this study for this reason. Finally, in order to provide a better understanding for the readers, we removed any information regarding Male and shaped it (please see Data source section in the Methods).

Authors indicated that all girls aged between 15 to 19 years were included in the study and those out of the range were excluded. How was consent obtained for females less than 18 years of age? Was accent sought from this age group in the primary data.

Author’s response: We used Ethiopian Demographic and Health Survey data, which was collected by the Ethiopian Central Statistical Agency, as we tried to explain in the preceding comment. As a result, since the data is secondary, there is no need to get consent from the participants concerning the data.

Results

There seem to be copious repetition of the results in the tables in the write up. Authors can consider stating just significant ones and leave the rest in the illustrated tables for readers to peruse.

Author’s response: We amended it throughout the paper per the reviewer comment.

Discussion

In-text referencing of CSA, 2016 should be properly done.

Author’s response: Amended it per the reviewer comment.

Can authors identify some study limitations and how their study design or approach sought to curtail the limitations.

Author’s response: Yes, when using primary data as the research analysis, researchers can find solutions to the limitations. However, this may not be the case when using secondary data.

Additional Revisions 

1. We revised the introduction using more recent sources.

2. We explained the method step by step.

3. We removed the Hosmer and Lemeshow tests from the method, as well as the corresponding findings, because in multilevel modeling, whether to evaluate whether the data is good or bad for the model, Tests of Heterogeneity is sufficient. 

4. To have a better knowledge of multilevel modeling model comparison and adequate checking, we added the "Model comparison, adequacy, and heterogeneity checking" section in the methods part. 

5. Since we previously displayed the odds ratio with the confidence interval for Z in the last table. Now, we therefore provided the odds ratio together with the corresponding confidence interval.

---

## [Decision Letter · Decision Letter 1]

4 Jul 2023

PONE-D-22-18542R1Correlates and prevalence associated with early childbearing among teenage girls in Ethiopia: A Multilevel AnalysisPLOS ONE

Dear Dr. Abebe

Thank you for submitting your manuscript to PLOS ONE. After careful consideration, we feel that it has merit but does not fully meet PLOS ONE’s publication criteria as it currently stands. Therefore, we invite you to submit a revised version of the manuscript that addresses the points raised during the review process.

We look forward to receiving your revised manuscript.

Kind regards,

Lebeza Alemu Tenaw

Academic Editor

PLOS ONE

Additional Editor Comments:

- There is no clear description why you are intended to this study.

- What are the gaps you have seen from the previous studies?

- What does it mean teenage pregnancy? if it is pregnancy before the age of 18 years, how did you talk about marriage before 18 considered as a significant factor?

- The other challenge in your result is you identified non modifiable factors like religion as significant factor, what will be your recommendation?

- The recommendation is not based on your results.

Reviewers' comments:

Reviewer's Responses to Questions

**Comments to the Author**

1. If the authors have adequately addressed your comments raised in a previous round of review and you feel that this manuscript is now acceptable for publication, you may indicate that here to bypass the “Comments to the Author” section, enter your conflict of interest statement in the “Confidential to Editor” section, and submit your "Accept" recommendation.

Reviewer #2: All comments have been addressed

Reviewer #3: (No Response)

2. Is the manuscript technically sound, and do the data support the conclusions?

Reviewer #2: Yes

Reviewer #3: Partly

3. Has the statistical analysis been performed appropriately and rigorously? 

Reviewer #2: Yes

Reviewer #3: Yes

4. Have the authors made all data underlying the findings in their manuscript fully available?

Reviewer #2: Yes

Reviewer #3: Yes

5. Is the manuscript presented in an intelligible fashion and written in standard English?

Reviewer #2: Yes

Reviewer #3: No

6. Review Comments to the Author

Reviewer #2: I commend the authors for revising the manuscript and effecting changes at the various points appropriately.

Reviewer #3: Thanks authors as they provided response to previous comments, however, there are some comments that should be addressed by authors as follows:

Abstract:

Methods: Please write about instrument that you used for data collection. For instance how authors assessed media used by participants.

Results: please provide the prevalence of early childbearing teenage as you stated in the title of manuscript.

Introduction

1. Readers like to know is child marriage legal in Ethiopia? What is the legal age for marriage?

2. Please provide some information about health care of adolescents who got married for for instance if they receive contraception and other cares.

Methods

1. As authors used data collection in 2016, readers would like to know the present situation of Ethiopia.

Results

1. Please provide the number of teenagers who were pregnant along with the percentage.

2. Please report the number and percentages of teenagers who were pregnant in age of <15 and between 15 and 19.

3. Page 16, there are two lines that are repetition of their earlier lines, please remove them.

4. As in Table 1, authors stated the age in first marriage, I would like to know, did you have second marriage in your study? How many were pregnant without marriage?

5. Table 1: did you ask about non-modern contraception?

6. The age difference between teenagers and their husbands may play a role in teenagers pregnancy, please report it along with their occupation.

7. All tables: please report p=0.000 as p<0.0001.

Discussion

1. In Comparison the rate of teenage pregnancy in the present study with other studies, indicate the year of the study. This is good to know if the teenage pregnancy is in decrease.

2. Please expand the results of the study regarding why Muslim teenagers have early pregnancy and more children?

7. PLOS authors have the option to publish the peer review history of their article (what does this mean?). If published, this will include your full peer review and any attached files.

Reviewer #2: No

Reviewer #3: No

---

## [Author Response · Author response to Decision Letter 1]

10 Jul 2023

PONE-D-22-18542R1

Correlates and prevalence associated with early childbearing among teenage girls in Ethiopia: A Multilevel Analysis

PLOS ONE

Title Modification: We interchanged the word “Correlates and prevalence” by “Prevalence and correlates” in the title.

Additional Editor Comments:

- There is no clear description why you are intended to this study.

Author’s response: We elaborated more on it per the editorial comment in the last paragraph of the introduction.

- What are the gaps you have seen from the previous studies?

Author’s response: We elaborated more on it per the editorial comment in the last paragraph of the introduction.

- What does it mean teenage pregnancy?

Author’s response: We defined it on the first two citations of the introduction of this study.

- If it is pregnancy before the age of 18 years, how did you talk about marriage before 18 considered as a significant factor?

Author’s response: Dear editor, our study is only based on teenage girls under the age of nineteen, as we have pointed out in the exclusion and exclusion criteria. Even though marriage under 18 is illegal in Ethiopia, Ethiopian demographic and Health survey data show that most Ethiopian girls marry under the age of 18. As a result, marriage before 18 can be a significant factor associated with teenage childbearing.

-Can we assess prevalence by Odds ratio?

Author’s response: No, we can assess only the correlates (the associated factors) of teenage childbearing by Odds ratio. We only used the prevalence in the title to show the prevalence rate of early teenage childbearing, and we added the prevalence in the result of the abstract.

-You have assessed the associated factors, so how can you know about the burden/ prevalence of teenage pregnancy?

Author’s response: We also included its prevalence, as stated in the study's title. We incorporated the prevalence rate in the abstract and reported it as 10.26%.

- There is no clear description about the sample size, procedure of extraction the participants.

Author’s response: Amended per the editor comments.

- The other challenge in your result is you identified non modifiable factors like religion as significant factor, what will be your recommendation?

Author’s response: We have tried to elaborate on the possible explanations in the discussion, and conclusion. 

- The recommendation is not based on your results.

Author’s response: We have tried to modify it.

Reviewers' comments:

Reviewer's Responses to Questions

Comments to the Author

1. If the authors have adequately addressed your comments raised in a previous round of review and you feel that this manuscript is now acceptable for publication, you may indicate that here to bypass the “Comments to the Author” section, enter your conflict of interest statement in the “Confidential to Editor” section, and submit your "Accept" recommendation.

Reviewer #2: All comments have been addressed

Reviewer #3: (No Response)

2. Is the manuscript technically sound, and do the data support the conclusions?

Reviewer #2: Yes

Reviewer #3: Partly

3. Has the statistical analysis been performed appropriately and rigorously?

Reviewer #2: Yes

Reviewer #3: Yes

4. Have the authors made all data underlying the findings in their manuscript fully available?

Reviewer #2: Yes

Reviewer #3: Yes

5. Is the manuscript presented in an intelligible fashion and written in standard English?

Reviewer #2: Yes

Reviewer #3: No

6. Review Comments to the Author

Reviewer #2: I commend the authors for revising the manuscript and effecting changes at the various points appropriately.

Reviewer #3: Thanks authors as they provided response to previous comments, however, there are some comments that should be addressed by authors as follows:

Abstract:

Methods: Please write about instrument that you used for data collection. For instance how authors assessed media used by participants.

Author’s response: We used data from the Ethiopian Demographic and Health Survey, gathered by the Ethiopian Central Statistical Agency. We considered mass media exposure as watching television, listening to the radio, and reading a newspaper or magazine at least once a week. We incorporated it in the study variable section.

Results: please provide the prevalence of early childbearing teenage as you stated in the title of manuscript.

Author’s response: We have provided it per the reviewer’s comment.

Introduction

1. Readers like to know is child marriage legal in Ethiopia? What is the legal age for marriage?

According to Ethiopia's amended family code, getting married before reaching 18 is no longer acceptable.

Author’s response: In the introduction's fifth paragraph, we elaborated on it.

2. Please provide some information about health care of adolescents who got married for instance if they receive contraception and other cares.

Author’s response: In the end of introduction's fourth paragraph, we tried to elaborate about it.

Methods

1. As authors used data collection in 2016, readers would like to know the present situation of Ethiopia.

Author’s response: Due to a deficient national documentation system and the fact that past studies were conducted on a local scale, institutionally or regionally focused, there is no evidence available on the prevalence of early teen pregnancy in Ethiopia nowadays.

Results

1) Please provide the number of teenagers who were pregnant along with the percentage.

Author’s response: We have provided it in the second sentence of descriptive result as well as row 2 of table 1.

2) Please report the number and percentages of teenagers who were pregnant in age of <15 and between 15 and 19.

Author’s response: We have provided it in the third sentence of descriptive result.

3) Page 16, there are two lines that are repetition of their earlier lines, please remove them.

Author’s response: Removed the repetition and reshaped it.

4) As in Table 1, authors stated the age in first marriage, I would like to know, did you have second marriage in your study? How many were pregnant without marriage?

Author’s response: The number of participants who were pregnant without marriage was not considered in our study.

5) Table 1: did you ask about non-modern contraception?

Author’s response: We did not gather the data ourselves. We used data from the Ethiopian Demographic and Health Survey, gathered by the Ethiopian Central Statistical Agency. They collected data on 16,583 women in total, but only 3,498 of those women provided all the necessary information. Also, they gathered data on non-modern methods of contraception, and the EDHS report showed that some respondents used them; however, when the missing data was cleared, none of the women who used non-modern methods of contraception were included.

6) The age difference between teenagers and their husbands may play a role in teenagers pregnancy, please report it along with their occupation.

Author’s response: We accepted this insightful recommendation.

However, we chose only eleven variables for consideration at the beginning of this study out of a huge number of variables in the Ethiopian Demographic and Health Survey data. Therefore, it is now challenging to include the husbands' age and work status at this point because doing so results in the responses of the women not being matched unless all variables are selected at the same time. In order to perform it now, it takes time. We hope that you can understand this problem.

7) All tables: please report p=0.000 as p<0.0001.

Author’s response: We amended it throughout the table per the reviewer comment.

Discussion

1. In Comparison the rate of teenage pregnancy in the present study with other studies, indicate the year of the study. This is good to know if the teenage pregnancy is in decrease.

Author’s response: We provided it in the document.

2. Please expand the results of the study regarding why Muslim teenagers have early pregnancy and more children?

Author’s response: We have tried to elaborate on the possible explanations in the discussion. ________________________________________

---

## [Editor Report · Decision Letter 2]

12 Jul 2023

Prevalence and correlates Associated with early childbearing among teenage girls in Ethiopia: A Multilevel Analysis

PONE-D-22-18542R2

Dear Dr. Abebe

We’re pleased to inform you that your manuscript has been judged scientifically suitable for publication and will be formally accepted for publication once it meets all outstanding technical requirements.

Kind regards,

Lebeza Alemu Tenaw

Academic Editor

PLOS ONE

---

## [Editor Report · Acceptance letter]

31 Jul 2023

PONE-D-22-18542R2 

Prevalence and Correlates Associated with early childbearing among teenage girls in Ethiopia: A Multilevel Analysis 

Dear Dr. Liga:

I'm pleased to inform you that your manuscript has been deemed suitable for publication in PLOS ONE. Congratulations! Your manuscript is now with our production department. 

Kind regards, 

on behalf of

Mr. Lebeza Alemu Tenaw 

Academic Editor

PLOS ONE